# Unique Severe COVID-19 Placental Signature Independent of Severity of Clinical Maternal Symptoms

**DOI:** 10.3390/v13081670

**Published:** 2021-08-23

**Authors:** Marjolein F. Husen, Lotte E. van der Meeren, Robert M. Verdijk, Pieter L. A. Fraaij, Annemiek A. van der Eijk, Marion P. G. Koopmans, Liv Freeman, Hein Bogers, Marjolijn D. Trietsch, Irwin K. M. Reiss, Philip L. J. DeKoninck, Sam Schoenmakers

**Affiliations:** 1Department of Obstetrics and Gynaecology, Erasmus MC, University Medical Center, 3015 GD Rotterdam, The Netherlands; m.f.husen@erasmusmc.nl (M.F.H.); p.dekoninck@erasmusmc.nl (P.L.J.D.); 2Department of Pathology, Leiden University Medical Center, 2333 ZA Leiden, The Netherlands; L.van_der_Meeren@lumc.nl; 3Department of Pathology, University Medical Center Utrecht, 3584 CX Utrecht, The Netherlands; 4Department of Pathology, Erasmus MC, University Medical Center, 3015 GD Rotterdam, The Netherlands; 5Department of Viroscience, Erasmus MC, University Medical Center, 3015 GD Rotterdam, The Netherlands; p.fraaij@erasmusmc.nl (P.L.A.F.); a.vandereijk@erasmusmc.nl (A.A.v.d.E.); 6Department of Pediatric Infectiology, Immunology and Rheumatology, Erasmus University Medical Center, 3015 GD Rotterdam, The Netherlands; m.koopmans@erasmusmc.nl; 7Department of Obstetrics and Gynaecology, Ikazia Hospital, 3083 AN Rotterdam, The Netherlands; L.freeman@ikazia.nl; 8Department of Obstetrics and Gynaecology, Franciscus en Vlietland Hospital, 3045 PM Rotterdam, The Netherlands; H.bogers@franciscus.nl; 9Department of Obstetrics and Gynaecology, Leiden University Medical Center, 2333 ZA Leiden, The Netherlands; M.D.Trietsch@lumc.nl; 10Department of Neonatology, Erasmus MC, University Medical Center, 3015 GD Rotterdam, The Netherlands; i.reiss@erasmusmc.nl

**Keywords:** placenta, SARS-CoV-2, pregnancy, foetal outcome

## Abstract

Background: Although the risk for transplacental transmission of SARS-CoV-2 is rare, placental infections with adverse functional consequences have been reported. This study aims to analyse histological placental findings in pregnancies complicated by SARS-CoV-2 infection and investigate its correlation with clinical symptoms and perinatal outcomes. We want to determine which pregnancies are at-risk to prevent adverse pregnancy outcomes related to COVID-19 in the future. Methods: A prospective, longitudinal, multicentre, cohort study. All pregnant women presenting between April 2020 and March 2021 with a nasopharyngeal RT-PCR-confirmed SARS-CoV-2 infection were included. Around delivery, maternal, foetal and placental PCR samples were collected. Placental pathology was correlated with clinical maternal characteristics of COVID-19. Results: Thirty-six patients were included, 33 singleton pregnancies (*n* = 33, 92%) and three twin pregnancies (*n* = 3, 8%). Twenty-four (62%) placentas showed at least one abnormality. Four placentas (4/39, 10%) showed placental staining positive for the presence of SARS-CoV-2 accompanied by a unique combination of diffuse, severe inflammatory placental changes with massive perivillous fibrin depositions, necrosis of syncytiotrophoblast, diffuse chronic intervillositis, and a specific, unprecedented CD20+ B-cell infiltration. This SARS-CoV-2 placental signature seems to correlate with foetal distress (75% vs. 15.6%, *p* = 0.007) but not with the severity of maternal COVID-19 disease. Conclusion: We describe a unique placental signature in pregnant patients with COVID-19, which has not been reported in a historical cohort. We show that the foetal environment can be seriously compromised by disruption of placental function due to local, devastating SARS-CoV-2 infection. Maternal clinical symptoms did not predict the severity of the SARS-CoV-2-related placental signature, resulting in a lack of adequate identification of maternal criteria for pregnancies at risk. Close foetal monitoring and pregnancy termination in case of foetal distress can prevent adverse pregnancy outcomes due to COVID-19 related placental disease.

## 1. Introduction

COVID-19 in pregnancy is associated with increased severe maternal morbidity and mortality as well as adverse pregnancy outcome [1,2,3]. Some studies suggest, perinatal risks and pregnancy outcomes appear to be related to the severity of the illness in women with COVID-19 [1,4,5]. Despite a limited risk of vertical transmission (with probable cases estimated to be around 3–4%), the prevalence of placental infection in women with COVID-19 reportedly is up to 7.7–21% [6,7] and its clinical consequences should be considered.

The physiological adaptations and dynamics of the maternal immune system during pregnancy ensure the maintenance of defence mechanisms for maternal and foetal survival while simultaneously creating an essential immune-tolerant environment preventing rejection of the semi-allogenic foetus. Successful initiation of pregnancy is characterized by a pro-inflammatory and antiviral profile, followed by the anti-inflammatory or T helper 2 (Th2) stage allowing foetal growth, and a switch back to T helper 1 (Th1) or pro-inflammatory state at the third trimester [8,9]. Due to the Th1/Th2 dynamics and pregnancy-related physiological adaptations, such as an oedematous upper respiratory tract and diaphragm elevation [10], pregnant women are generally considered to be more susceptible to viral respiratory pathogens compared to non-pregnant women [11]. To protect the developing foetus, the placenta functions both as a physical as well as an immunological barrier between the maternal and foetal compartment [12]. However, the local immune tolerant environment, at the level of the placenta, allows for some susceptibility to viral replication and transplacental transmission, as has been shown with TORCH infections (toxoplasmosis, “other”, rubella, CMV and herpes simplex) [13] and more recently with the Zika virus [14,15]. Other general routes of placental infection include regions where the integrity of the syncytiotrophoblast is damaged due to hemodynamic shear stress, immune-mediated or hypoxic injury resulting in fibrin degradation compromising the feto-maternal barrier [16].

After transmission, SARS-CoV-2 replicates in either the respiratory or gastrointestinal tract resulting in a wide spectrum of diseases ranging from very mild to severe respiratory symptoms [17,18]. The severity of COVID-19, in general, seems to be associated with higher SARS-CoV-2 viral loads and plasma viremia [19]. Still, despite this finding maternal viremia or placental infection was initially thought not to be a major issue as published in a prospective cohort study in 2020 [20]. However, a recent study in symptomatic pregnant women admitted at the hospital for COVID-19 related symptoms, showed that maternal viremia was associated with SARS-CoV-2 presence in the foetal compartment [21], hence indicative of transplacental transmission of SARS-CoV-2. Maternal viremia seems imperative to allow placental infection by SARS-CoV-2. For the SARS-CoV-2 virus, its primary receptor is the membrane-bound Angiotensin-Converting Enzyme 2 (ACE2), which is expressed in the human placenta, throughout all trimesters [22]. The presence of ACE2-receptor allows for a possible direct entrance for SARS-CoV-2 particles from the maternal circulation into the placental tissue, which in combination with the placental immunotolerance, could result in a high risk for placental infection.

The current study aims to determine if placental infection by SARS-CoV-2 can be predicted based on the severity of maternal clinical presentation of COVID-19 and to identify pregnancy at-risk for adverse foetal outcomes. The reported SARS-CoV-2 associated histological changes in the placenta are chronic intervillositis, chronic villitis, massive perivillous fibrin depositions and syncytiotrophoblast damage resulting in necrosis [7,20,23,24,25,26,27,28,29], which can limit the essential gas- and nutrient exchange at the maternal-foetal interface, necessary for foetal survival [24]. The remaining functional placental interface relative to the amount of damaged or fibrin-occluded interface will determine the level of effect on foetal growth and development. As a rule, the above-mentioned histological findings are associated with a high risk of adverse perinatal outcomes [30], but they are only diagnosed in retrospect and thus cannot be used for clinical decision-making. However, if it is possible to determine which pregnancies are at-risk for placental SARS-CoV-2 infection, it would allow detection and prevention of adverse pregnancy outcomes related to COVID-19 in the future.

Therefore, this study first describes the histological placental findings in pregnancies complicated by either an active or resolved SARS-CoV-2 infection. Secondly, we examined whether the severity of maternal presentation of COVID-19 is related to the degree of histological placental changes and the perinatal outcome.

## 2. Materials and Methods

### 2.1. Study Design and Patients

The current study was designed as a prospective, longitudinal, multicentre (Erasmus MC, University Medical Center, Rotterdam; Leiden University Medical Center, Leiden; Ikazia, Rotterdam; Franciscus Gasthuis & Vlietland, Rotterdam, all in The Netherlands), cohort study. All pregnant women presenting between April 2020 and March 2021 with SARS-CoV-2 (confirmed by positive PCR for SARS-CoV-2 on nasopharyngeal samples) were included. Patients were screened for COVID-19 in case they had clinical symptoms or in case of a high-risk contact.

For each individual case, we aimed to collect the following samples for SARS-CoV-2 RT-PCR: vagina/posterior fornix (swab), urine and occasionally faeces. The onset, duration and severity of disease symptoms, hospital admission, or home confinement, was documented for every patient. We defined three categories for the severity of the clinical symptoms based upon outcome measure set of patient state: ambulatory mild disease, asymptomatic and symptomatic), hospitalized: moderate (no respiratory failure), hospitalized: severe (respiratory failure and in need for therapy, e.g., oxygen or conventional ventilation) [31]. Maternal, foetal and neonatal characteristics were retrospectively retrieved from the electronic patient records. Low dose aspirin was not routinely prescribed for COVID-19 in our study population.

To study and allow an adequate evaluation of the risk for possible vertical transmission, additional swabs of the vagina, both the maternal and foetal side of the placenta, and the neonatal oropharynx were collected directly after birth, when possible. Furthermore, maternal blood and umbilical cord samples were analysed with RT-PCR for SARS-CoV-2 RNA-emia and SARS-CoV-2 specific antibodies (immunoglobulins G and M). All above-mentioned sampling were standard of care according to the local SARS-CoV-2 during pregnancy protocol.

The study population was divided into two subgroups at the time of delivery: pregnant women with an active SARS-CoV-2 and pregnant women who had recovered from SARS-CoV-2 with a confirmed positive nasopharyngeal SARS-CoV-2 PCR at any moment in pregnancy. To evaluate the possible association of placental histopathological findings with maternal characteristics of COVID-19 based upon the three defined categories [32], placentas with positive SARS-CoV-2 immunohistochemistry versus placenta with negative SARS-CoV-2 immunohistochemistry were compared. In accordance with local protocol, placentas of uncomplicated term pregnancies are not sent for pathological examination. Therefore, findings of a historical cohort of placentas of uncomplicated, healthy term pregnancies of vaginal delivery (*n* = 477) and elective caesarean sections (*n* = 105), prior to the first reported case of COVID-19 in China, were used as controls for comparison [30,33].

The study was reviewed and received approval from the local institutional Medical Ethics Committee of the Erasmus MC University Medical Center, Rotterdam according to the Dutch Medical Research with Human Subjects Law (MEC-2020-0323). Written informed consent was obtained from each enrolled patient, including informed consent for neonatal information collection. Patients were excluded from the current study if there was an inability to give informed consent.

### 2.2. Virology

Real-time quantitative PCR (RT-qPCR) was performed for the detection of SARS-CoV-2. Swabs were collected in tubes containing 4 mL virus transport medium (Dulbecco’s modified eagle’s medium (DMEM, Lonza, Switzerland) supplemented with 40% FBS, 20 mM 4-(2-hydroxyethyl)-1-1piperazineethanesulfonic acid (HEPES), NaCO_3_, 10 μg/mL amphotericin B, 1000 μg/mL streptomycin). Assays were performed directly on urine, faeces and EDTA blood [34].

Wantai SARS-CoV-2 total Ig and IgM ELISAs (Beijing Wantai Biological Pharmacy Enterprise Co., Ltd., Beijing, China) was performed according to the manufacturer’s protocol. The ELISAs were coated with RBD antigen [35].

### 2.3. Placenta Histopathology

All placentas were examined independently by two experienced perinatal pathologists. The macroscopic examination included the documentation of placental weight and the presence of parenchymal abnormalities. Samples were taken from the umbilical cord, membranes, parenchyma and macroscopic lesions. Formalin-fixed, paraffin-embedded tissues were H&E stained according to standardized international protocol [36]. All cases were reviewed according to the international guideline for placenta evaluation, Amsterdam Placental Consensus Statement [36], which in short includes evaluation for pathological findings resulting in placental dysfunction [37] and inflammatory changes in the maternal and foetal compartments [32,38,39].

### 2.4. Immunohistochemical Stainings for Immune Subsets (CD3, CD20, CD68)

Local inflammatory responses were further evaluated by diagnostic immunohistochemical markers CD3 (mono-clonal anti-Rabbit, clone 3VG6, Roche, Basel, Switzerland), CD20 (mono-clonal anti-Mouse, clone L2G, Roche, Basel, Switzerland) and CD68 (mono-clonal anti-Mouse, clone KP1, Roche, Basel, Switzerland) to visualise the location of respective T-cells, B-cells and macrophages. Sequential 4 µm thick (FFPE) sections were stained for CD3, CD20 and CD68 (all Ventana, Oro Valley, AZ, USA). In brief, following deparaffinisation and heat-induced antigen retrieval the tissue samples were incubated according to their optimized time and protocol with the antibody of interest according to the manufactures instructions (Ventana, Oro Valley, AZ, USA).

### 2.5. Immunohistochemistry of SARS-CoV-2 NP and Spike

The presence of SARS-CoV-2 antigen in the syncytiotrophoblast was assessed by immunohistochemistry polyclonal anti-rabbit SARS-CoV-2 NP (Sino-Biologicals, Beijing, China) and anti-rabbit SARS-CoV-2 Spike (Sabbiotech, College Park, MD, USA). To determine the presence of SARS-CoV-2 by automated IHC using the Discovery ULTRA (Ventana Medical Systems Inc., Oro Valley, AZ, USA). Tissue blocks were cut into 4 µm thick (FFPE) sections and were stained for SARS-CoV-2 Nucleoprotein and Spike protein. In brief, following deparaffinisation and heat-induced antigen retrieval with CC1 (#950-500, Ventana) for 32 min the tissue samples were incubated with the antibody for 32 min at 37 °C. Detection and incubation were done according to the manufactures instructions (Ventana, Oro Valley, AZ, USA). Positive controls were used on every slide.

### 2.6. Statistics

SPSS Statistics (IBM SPSS Statistics for Windows, version 25.0, Armonk, NY, USA) was used for statistical analysis. Categorical data were presented as numbers and percentages, normally distributed continuous data as mean (±SD) and non-normally distributed as median (IQR). Determination of differences between groups was calculated with independent sample t-tests for normally distributed continuous data, Mann–Whitney U test for non-normally distributed continuous data and χ^2^ test and Fisher’s exact test for categorical data. A *p*-value < 0.05 was considered statistically significant.

## 3. Results

### 3.1. Maternal Characteristics, Obstetric and Neonatal Outcomes

During the study period, 36 patients were included. The majority had a singleton pregnancy (*n* = 33, 92%), while the remaining patients had a twin pregnancy (*n* = 3, 8%). Table 1 shows the general maternal characteristics and obstetric outcomes for the total study population as well as the two subgroups: women with COVID-19 and women who had recovered from COVID-19 at the time of giving birth. The majority of patients (*n* = 28, 78%) reported only mild or no symptoms and 8 patients (22%) had either moderate or severe disease and were admitted to a dedicated isolation unit at the maternity ward due to COVID19-related complaints. All patients were managed by a multidisciplinary team of obstetricians, internal medicine and intensive care medicine specialists and neonatologists. Of the patients who needed hospitalisation the majority (*n* = 7, 88%) needed respiratory support at some stage. Five patients (*n* = 5, 63%) developed severe disease and required intensive care unit admission for intubation and mechanical ventilation. Four of these five patients (80%) were admitted to the ICU during the third trimester and one (20%) during the second trimester.

Of the 36 patients, 11 pregnancies (30.6%) resulted in preterm delivery. Pregnant women with active SARS-CoV-2 as compared to a previous SARS-CoV-2 were more likely to have preterm birth (respectively 52.9% vs. 10.5%, *p* = 0.01). Almost all preterm births (*n* = 10, 90.9%) were due to an emergency caesarean section. Causes for the emergency caesarean section were increasing maternal respiratory insufficiency (*n* = 4 (40%) and foetal distress (*n* = 6, 60%). All neonates (*n* = 39 were born alive. Of the neonates, 5 (12.8%) had an Apgar score <7 after 5 min; 3 of these 5 were after a preterm emergency caesarean section because of foetal distress. All 5 were born from mothers with active SARS-CoV-2 at the time of giving birth. Besides the higher incidence of an Apgar score <7 in patients with active COVID-19, the birthweight percentile and pH also appear to be lower in this group, yet the number of cases is too small to draw firm conclusions. Table 1 and Table 2 show an overview of the obstetric outcomes and neonatal characteristics in our study population, divided into two subgroups.

### 3.2. Placental Histopathological Examination

In all cases, the placenta was histologically examined (*n* = 39), only 15 (39%) placentas showed no placental abnormalities. Twenty-four (62%) placentas showed at least one abnormality, either sign of foetal vascular malperfusion (4/39), multifocal low-grade villitis of unknown aetiology (4/39), chorioamnionitis (6/39) and/or maternal vascular malperfusion (5/39). In six cases (15%) we observed signs of a resolved inflammation with a sporadic CD3+ lymphocyte in the villi without an active chronic villitis or intervillositis. In six cases (15%) nucleated red blood cells were present in the foetal circulation as a sign of foetal hypoxia.

Four placentas (10%) showed diffuse macroscopic visible white lesions that correlated with histological abnormalities of the placenta parenchyma. Histology showed a chronic intervillositis with diffuse CD68+ macrophages and CD3+ T-lymphocytes in the maternal compartment in combination with massive perivillous fibrin depositions and necrotic syncytiotrophoblast accompanied with multifocal low-grade or high-grade chronic villitis (Figure 1A). Remarkably, these four SARS-CoV-2 positive cases showed a distinctive CD20+ B-cell infiltrate in the maternal compartment which has never been described before (Figure 1B). Immunohistochemistry for SARS-CoV-2 showed positivity in the vital syncytiotrophoblast (Figure 1C,D), while villi with necrotic syncytiotrophoblast showed weak to no staining for SARS-CoV-2. A fifth placenta with inconclusive immunohistochemistry for SARS-CoV-2 showed similar histologic observations, including the specific CD20+ B-cell infiltration and diffuse necrotic changes with sporadically vital trophoblast tissue. Table 3 shows the placental outcomes, divided into two subgroups. There were no differences in presence of chorioamnionitis, chronic villitis and chronic intervillositis between the women who had active COVID-19 and recovered from SARS-CoV-2 at the time of giving birth.

A comparison of maternal clinical characteristics was made between patients with and without positive SARS-CoV-2 immunohistochemistry (Table 4). Besides, a significantly higher percentage of (gestational) diabetes in the group with a placental presence of SARS-CoV-2 (50% versus 12.5%, *p* = 0.02), no significant differences in maternal factors were observed that contributed to the severe inflammatory placental changes. Table 5 shows the placental characteristics, divided into two subgroups of patients with and without positive SARS-CoV-2 immunohistochemistry.

### 3.3. Clinical Presentation of COVID-19 in Cases of Confirmed Placental SARS-CoV-2

Of the four cases with a confirmed placental SARS-CoV-2, three patients had ambulatory symptomatic mild disease, while one patient was asymptomatic. The maternal infection of the four confirmed placental SARS-CoV-2 cases were all closer to the day of delivery than the mothers with a resolved COVID-19 infection (median 2.5 days vs. 21 days). For a comprehensive overview of the clinical presentation and neonatal outcome, see Appendix A.

The SARS-CoV-2 placental signature seems to correlate with foetal distress (75% vs. 15.6%, *p* = 0.007) but not with severity of maternal COVID-19 disease, resulting in a lack of adequate identification criteria for pregnancies at-risk. However, despite a lack of maternal severe disease, 50% (*n* = 2) of the women showed signs of systemic SARS-CoV-2 presence based on maternal viremia and vaginal fornix SARS-CoV-2 positivity (Table 4). All cases hospitalized for severe disease and an associated increased risk for maternal viremia [19,21], had negative SARS-CoV-2 PCRs in maternal blood or fornix posterior, showed negative immunohistochemical staining for SARS-CoV-2 of the placenta and did not have severe inflammatory abnormalities in the placenta.

## 4. Discussion

In our cohort, placental SARS-CoV-2 positivity was characterised by an unprecedented combination of chronic intervillositis, massive perivillous fibrin depositions, necrotic syncytiotrophoblast, chronic villitis and the presence of a unique, distinctive CD20+ B-cell infiltrate (Figure 1). Although some articles have mentioned sporadic B-cells, primarily in foetal vessels or in the stroma of villous stromal cells [40,41,42,43], a chronic intervillositis or chronic villitis with a CD20+ B-cell infiltrate has not been described in association with any other placental infection and the presence of B-cells is highly unusual in the placenta.

We compared our findings to a previously published Dutch cohort of placentas of uncomplicated healthy term pregnancies (*n* = 583) and the above-mentioned combination was not observed in these cohorts (Table 3, Table 4 and Table 5) [30,33]. Histological examination of these healthy control placentas did show the presence of a wide variety of histological abnormalities (even in absence of disease) ranging from signs of ischemia, delayed maturation to chronic villitis (Table 3, Table 4 and Table 5) [33]. In line with these findings, these non-specific histological placental abnormalities were also observed in two-thirds of cases (24/36) in our cohort (Table 3). However, additional placental abnormalities found in the uncomplicated pregnancies cohort due to other (maternal) factors in cases of placental SARS-CoV-2 infection might worsen placenta function. In the group with multiple placental lesions and immunohistochemical SARS-CoV-2 positive placentas, a significantly higher percentage of women had (gestational) diabetes, although the total number of patients is low. In general, diabetic women have a higher risk of placental dysfunction, which could predispose them to severe histopathological changes in case of placental SARS-CoV-2 infection [33,44]. As such, one can consider close monitoring of foetal well-being in women with gestational diabetes and COVID-19 during pregnancy. More studies on this subject are needed to define clinical care parameters for this combination.

The combination of severe placental abnormalities of chronic intervillositis, massive perivillous fibrin depositions, necrotic syncytiotrophoblast and the presence of a unique, distinctive CD20+ B-cell infiltrate seems not to be associated with maternal severity of COVID-19.

All four SARS-CoV-2 positive placentas show the same placental signature, with the specific presence of CD20+ B-cells in the maternal compartment, indicating a pathognomonic feature for placental SARS-CoV-2 infection. Observation of this placental signature has recently led to the identification of several women with a previously undiagnosed COVID-19 in their history (personal communication).

Our series provides evidence that placental SARS-CoV-2 infection can result in placental damage, thereby triggering subsequent local inflammation processes that can cause severe disruption of the placental structure and maternal-foetal interface. This translates into a reduced capacity of the fetoplacental unit to adapt to episodes of reduced placental blood flow, for instance during uterine contractions. This supports the hypothesis that the increased risk of foetal compromise is not related to vertical transmission but due to functional impairment of the placenta caused by severe placental infection as was recently described [45]. Lower birthweight, Apgar score and pH in active COVID-19 cases add further evidence that COVID-19 increases the risk of foetal compromise.

To our knowledge, this is the first study to describe a unique placental signature in multiple patients with COVID-19 without a severe maternal clinical presentation. The main strength is that we were able to correlate placental findings with maternal systemic SARS-CoV-2 presence, including maternal viremia and neonatal clinical outcomes. Currently, there are no treatment options directed against active SARS-CoV-2 virus particles. Based on these findings, it will be possible to determine pregnancies at-risk for placental SARS-CoV-2 infection and closely monitor these pregnancies to prevent adverse pregnancy outcomes related to COVID-19. In addition, we used a standardised sampling protocol and placenta evaluation. The limitations of the study are the small sample size of our confirmed placental SARS-CoV-2 infection-cohort, the risk of selection bias and the lack of gestational age- and time period matched healthy placental control group. The selection of the study population was affected by whether patients signal SARS-CoV-2 symptoms and presented themselves for testing. Due to the national midwifery system in the Netherlands, where women can choose their location of delivery, either at home, a birth centre or a hospital, we cannot exclude an underrepresentation of mild cases.

## 5. Conclusions

We described a unique placental signature with a specific presence of CD20+ B-cells clustering around the necrotic syncytiotrophoblast combined with chronic intervillositis and perivillous fibrin depositions, in pregnant patients with COVID-19 not seen in historic controls. This signature did not correlate with the severity of disease but may be associated with maternal systemic SARS-CoV-2 presence (2/4 cases). We assume foetal well-being can potentially be threatened by an increased risk of histopathologic placental abnormalities resulting from a localised immunological response triggered by the presence of SARS-CoV-2 in the placenta. Therefore, in the case of COVID-19 in pregnancy, we recommend SARS-CoV-2 PCR of maternal blood and vaginal fornix, to identify women at risk of placental infection with SARS-CoV-2 and to prevent negative pregnancy outcomes due to COVID-19, both for the mother as for her offspring. We also advocate, besides, the SARS-CoV-2 PCRs of blood and vagina, for histological examination of the placenta with a specific request for SARS-CoV-2 and CD20+ staining when confronted with an unexpected, adverse pregnancy outcome during the current COVID-19 pandemic. Revealing the cause of an adverse pregnancy outcome will impact obstetrical counselling and management of future pregnancies to prevent future foetal problems due to COVID-19 in pregnancy.

## Figures and Tables

**Figure 1 viruses-13-01670-f001:**
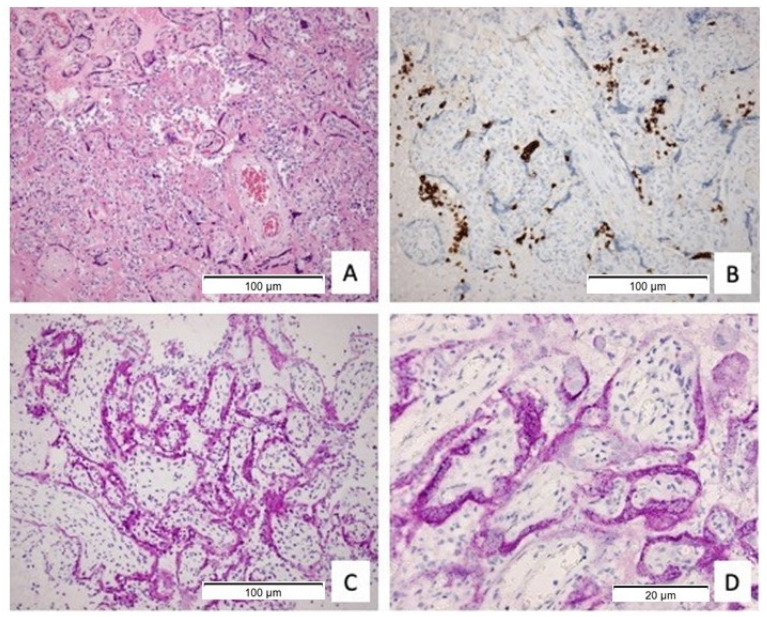
(**A**) histopathology of the placenta (HE staining) diffuse perivillous fibrin, necrotic syncytiotrophoblast and a chronic intervillositis (100×); (**B**) intervillous infiltrate of CD20+ B-cell lymphocytes (100×); (**C**) immunohistochemical staining for SARS-CoV-2 protein-specific antibody in the syncytiotrophoblast (100×); (**D**) immunohistochemical staining for SARS-CoV-2 spike protein-specific antibody localizing to the cytoplasm (400×).

**Table 1 viruses-13-01670-t001:** General maternal characteristics and obstetric outcomes for the total study population and subgroups active COVID-19 and resolved COVID-19 at time of giving birth.

	Total (*n* = 36)	Active COVID-19 (*n* = 17)	Resolved COVID-19 (*n* = 19)	*p*-Value
Age, years, mean (±SD)	33.5 (±5.1)	34.4 (±4.5)	32.8 (±5.7)	0.38
Primigravida, *n* = 35 (%)	9 (25.0)	6 (35.5)	3 (15.8)	0.21
Pre-pregnancy BMI, kg/m^2^, *n* = 32, median (IQR)	28.8 (10.4)	31.6 (12.5)	26.4 (9.5)	0.12
Geographic origin, *n* (%)				
Caucasian	10 (27.8)	4 (23.5)	6 (31.6)	0.59
North African	5 (13.9)	2 (11.8)	3 (15.8)	0.73
Turkish	4 (11.1)	2 (11.8)	2 (10.5)	0.91
Other African	3 (8.3)	1 (5.9)	2 (10.5)	0.62
Hispanic	2 (5.6)	1 (5.9)	1 (5.3)	0.94
Asian	2 (5.6	1 (5.9)	1 (5.3)	0.94
Other	6 (16.7)	4 (23.5)	2 (10.5)	0.3
Unknown	4 (11.1)	2 (11.8)	2 (10.5)	0.91
Comorbidities, *n* = 34 (%)				
Autoimmune disease	3 (8.3)	1 (5.9)	2 (10.5)	0.88
Pulmonary disease	0 (0.0)	-	-	-
(Gestational) Diabetes	6 (16.7)	5 (29.4)	1 (5.3)	0.15
Hypertensive disorders	2 (5.6)	0 (0.0)	2 (10.5)	0.39
Periconceptional smoking, *n* = 25 (%)	3 (8.3)	2 (11.8)	1 (5.3)	0.68
Singleton pregnancy, *n* (%)	33 (91.7)	16 (94.1)	17 (89.5)	0.62
GA (in days) at diagnosis, median (IQR)	237.5 (58)	234 (50)	241 (53)	0.05
Trimester at diagnosis, *n* (%)				
First trimester	1 (2.8)	0 (0)	1 (5.3)	0.34
Second trimester	4 (11.1)	1 (5.9)	3 (15.8)	0.35
Third trimester	31 (86.1)	16 (94.1)	15 (78.9)	0.19
Admission during pregnancy, *n* (%)	14 (38.9)	9 (52.3)	5 (26.3)	0.1
COVID-19 related	8 (22.2)	5 (29.4)	3 (15.8))	0.33
Respiratory failure	*7 (19.4)*	*5 (29.4)*	*2 (10.5)*	
Vomiting	*1 (2.8)*	*0 (0)*	*1 (5.3)*	
Non-COVID-19 related	6 (16.7)	4 (23.5)	2 (10.5)	0.3
GA at delivery (days) median (IQR)	269.5 (39)	243 (48)	271 (9)	0.04
Confirmed diagnosis to delivery (days) median (IQR)	19.5 (33)	4 (7)	35 (52)	0.00
Preterm delivery, *n* (%)	11 (30.6)	9 (52.9)	2 (10.5)	0.01
Spontaneous labour	1 (2.8)	0 (0)	1 (5.3)	0.03
Labour by induction	10 (27.8)	9 (52.9)	1 (5.3)	0.03
Mode of delivery, *n* (%)				
Vaginal spontaneous	13 (36.1)	4 (23.5)	9 (47.4)	0.14
Vaginal instrumental	1 (2.8)	0 (0)	1 (5.3)	0.34
Caesarean Section	22 (61.2)	13 (76.5)	9 (47.4)	0.07
Type of caesarean section, *n* (%)				
Elective	10 (27.8)	3 (17.6)	8 (42.1)	0.19
Emergency	12 (32.3)	10 (58.8)	2 (10.5)	0.002
Indication for caesarean section, *n* (%)				
Elective				
Breech position	3 (8.3)	1 (5.9)	2 (10.5)	0.33
Repeat caesarean section	4 (11.1)	2 (11.8)	2 (10.5)	0.68
Other	2 (5.6)	0 (0)	2 (10.5)	0.08
Unknown	1 (2.8)	0 (0)	1 (5.3)	0.22
Emergency				
Foetal distress	7 (19.4)	5 (29.4)	2 (10.5)	0.42
Maternal condition	5 (13.9)	5 (29.4)	0 (0)	0.03

Data are presented as mean and standard deviation (SD), median and interquartile range (IQR) or number (*n*) and percentage (%). BMI, body mass index in kilograms/square meter; GA, gestational age; *n*, number; %, percentage.

**Table 2 viruses-13-01670-t002:** Neonatal outcomes for the total study population and subgroups active COVID-19 and resolved COVID-19 at the time of giving birth.

	Total (*n* = 39)	Active COVID-19 (*n* = 18)	Resolved COVID-19 (*n* = 21)	*p*-Value
Birthweight percentile for GA, median (IQR)	48.0 (48)	21.5 (69)	51 (38)	0.22
SGA (BW < 10th percentile)	9 (23.1)	6 (33.3)	3 (14.2)	0.26
Apgar <7 at 5 min, *n* (%)	5 (12.8)	5 (29.4)	0 (0)	0.03
pH < 7.10, *n* (%)	2 (5.1)	2 (11.8)	0 (0)	0.17
PCR nasopharynx SARS-CoV-2 positive (23/39)	0	-	-	-
PCR umbilical cord blood SARS-CoV-2 positive (12/39)	0	-	-	-
Ig neonatal SARS-CoV-2 positive (1/39)	1 (2.6)	0 (0.0)	1 (5.3)	
Ig umbilical cord blood SARS-CoV-2 positive (14/39)	6 (15.4)	0 (0.0)	6 (28.6)	0.001

Data are presented as the median and interquartile range (IQR) or number (*n*) and percentage (%). GA, gestational age; *n*, number; %, percentage.

**Table 3 viruses-13-01670-t003:** Placenta outcomes for the total study population and subgroups active COVID-19 and resolved COVID-19 at the time of giving birth.

	Total (*n* = 39)	Active COVID-19 (*n* = 18)	Resolved COVID-19 (*n* = 21)	*p*-Value	Historical Healthy Term Controls, *n* = 583 [30,33]
Normal maturation (%)	29 (74.4)	9 (50.0)	20 (95.2)	0.001	15–20%
Delayed maturation (%)	10 (26.6)	9 (50.0)	1 (4.5)		
Chorioamnionitis, *n* (%)	10 (25.6)	3 (16.7)	7 (33.3)	0.24	5–25%
Chronic membranitis, *n* (%)	1 (2.6)	1 (5.6)	0 (0)	0.27	8–19% (term)
Chronic deciduitis, *n* (%)	6 (15.4)	3 (16.7)	3 (14.3)	0.84	2–13% (term)
Chronic villitis, *n* (%)	10 (25.6)	5 (27.8)	5 (23.8)	0.78	2–25%
Resolved villitis, *n* (%)	6 (15.4)	2 (11.1)	4 (19.0)	0.49	5–15%
Chronic intervillositis, *n* (%)	6 (15.4)	3 (16.7)	3 (14.3)	0.84	0.4%
Positive SARS IHC, *n* (%)	4 (10.3)	3 (16.7)	1 (4.8)	0.22	Not present
Perivillous fibrin	9 (23.1)	6 (33.3)	3 (14.3)	0.16	0.028% to 0.5%
Fibrinoid Necrosis	4 (10.3)	3 (16.7)	1 (4.8)	0.22	Not present
Maternal CD20+ B-cell infiltrate	5 (12.8)	3 (16.7)	2 (9.5)	0.74	Not present
PCR maternal side positive	2 (5.1)	2 (11.1)	0 (0)	0.12	-
PCR foetal side positive	2 (5.1)	2 (11.1)	0 (0)	0.12	-

Data are presented as number (*n*) and percentage (%). IHC, immunohistochemistry; PCR, polymerase chain reaction test.

**Table 4 viruses-13-01670-t004:** Maternal characteristics for the total study population and subgroups positive and negative SARS-CoV-2 immunohistochemical placentas.

	Total (*n* = 36)	SARS-CoV-2 Immunohistochemical Positive Placenta (*n* = 4)	SARS-CoV2 Immunohistochemical Negative Placenta (*n* = 32)	*p*-Value
Age, years, mean (±SD)	33.5 (±5.1)	31.5 (6.2)	33.8 (5.0)	0.40
Pre-pregnancy BMI, kg/m^2^, median (IQR)	28.8 (10.4)	30.1	28.2 (10.1)	1.00
Missing (*n* = 4)				
Geographic origin, *n* (%)				
Caucasian	10 (27.8)	0 (0.0)	10 (31.2)	0.19
Other	26 (72.2)	4 (100.0)	22 (68.8)	0.19
Comorbidities (%)				
Missing (*n* = 2)				
Auto immune disease	3 (8.3)	0 (0.0)	3 (9.8)	0.17
Pulmonary disease	0 (0.0)	-	-	-
(Gestational) Diabetes	6 (16.7)	2 (50.0)	4 (12.5)	0.02
Hypertensive disorders	2 (5.6)	0 (0.0)	2 (6.3)	0.18
GA (in days) at diagnosis, median (IQR)	237.5 (58)	226.5 (34)	243 (58)	0.98
Confirmed diagnosis to delivery (days) median (IQR)	19.5 (33)	2.5 (18)	21 (51)	0.06
Admission during pregnancy, *n* (%)	14 (38.9)	2 (50.0)	12 (37.5)	0.63
COVID-19 related	8 (22.2)	0 (0.0)	8 (25.0)	0.26
Non-COVID-19 related	6 (16.7)	2 (50.0)	4 (12.5)	0.06
Maternal PCR blood (*n* = 15)	1 positive (2.8)	1 (25.0)	0 (0.0)	0.000
Maternal PCR fornix (*n* = 25)	2 positive (5.6)	2 (50.0)	0 (0.0)	0.000
Signs of foetal distress, *n* (%)	8 (22.2)	3 (75.0)	5 (15.6)	0.007

Data are presented as mean and standard deviation (SD), median and interquartile range (IQR) or number (*n*) and percentage (%). GA, gestational age; *n*, number; %, percentage.

**Table 5 viruses-13-01670-t005:** Placental characteristics for the total study population and subgroups positive and negative SARS-CoV-2 immunohistochemical placentas.

	Total (*n* = 39)	SARS-CoV-2 Immunohistochemical Positive Placenta (*n* = 4)	SARS-CoV2 Immunohistochemical Negative Placenta (*n* = 35)		Historical Healthy Term Controls, *n* = 583 [30,33]
Perivillous fibrin, *n* (%)	9 (23.1)	4 (100.0)	5 (14.3)	0.000	0.028–0.5%
Fibrinoid Necrosis, *n* (%)	4 (10.3)	4 (100.0)	0 (0.0)	0.000	Not present
Maternal CD20+ B-cell infiltrate, *n* (%)	5 (12.8)	4 (100.0)	1 (2.9)	0.000	Not present
PCR maternal side positive (*n* = 30)	2 positive (5.1)	2 (50.0)	0 (0.0)	0.000	
PCR foetal side positive (*n* = 29)	2 positive (5.1)	2 (50.0)	0 (0.0)	0.000	

Data are presented in number (*n*) and percentage (%). *n*, number; %, percentage.

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
