# Peer review of "Unique Severe COVID-19 Placental Signature Independent of Severity of Clinical Maternal Symptoms"

_viruses, 2021, doi:10.3390/v13081670_

Round 1

Reviewer 1 Report

The paper is especially interesting even though the methods and study design from the onset were not exceptionally sound. It is still a very valuable set of data. There could be more emphasis on several very important findings including:  

62% of placentas examined had pathologic findings following maternal COVID-19 infection during pregnancy. Active maternal infection can be associated with placental infection and impaired fetal well being, even when mothers are asymptomatic or have mild symptoms

SARS-CoV2 placental infection and pathologic signature correlates with fetal distress (75% vs. 15%).

But maternal illness severity is not predictive of placental or fetal outcomes.   

Active COVID-19 infection increases the risk of prematurity (52%), predominantly by induction due to concerns for fetal well being

Active COVID-19 infection, specifically those with placental infection is associated with placental pathology, lower infant BW, APGAR, pH especially in women with obesity? and GDM. Case review demonstrates the risk is high in the first few days after the onset of infection. Thereafter, placental pathology may increase the risk of IUGR

Findings suggest an indirect relationship of SARS-CoV2 IgG in cord blood and neonatal outcomese (low APGAR, pH and BW) suggesting maternal antibodies may be protective

See detailed suggestions attached. 

Reviewer 2 Report

Thanks very much to the Authors to take a closer look on placentas, morphologically and after immunostain in pregnancies complicated by SARS-CoV-2 infections in correlation with the severity of maternal symptoms. I totally agree, that this may contribute to a better estimation of fetal risk.

Yet I would like to suggest few, very little changes:

  1. simplify and shorten the title (e.g. 'Is Placental signature independent of clinical maternal symptoms in COVID-19 infection?')
  2. many wordy tables; recommend to merge e.g. tab. 1 and 2 and probably 5, and tab. 4 and 6;
  3. Fig. 1: would like to recommend to adjust contrast and sharpen, and agree on upper- or lower case letters in photos and legend;
